# Overview of Systematic Reviews of Health Interventions for the Prevention and Treatment of Overweight and Obesity in Children

**DOI:** 10.3390/nu15030773

**Published:** 2023-02-02

**Authors:** Edgar Denova-Gutiérrez, Alejandra González-Rocha, Lucía Méndez-Sánchez, Berenice Araiza-Nava, Nydia Balderas, Giovanna López, Lizbeth Tolentino-Mayo, Alejandra Jauregui, Lucia Hernández, Claudia Unikel, Anabelle Bonvecchio, Teresa Shamah, Simón Barquera, Juan A. Rivera

**Affiliations:** 1Nutrition and Health Research Center, National Institute of Public Health, Cuernavaca 62100, Mexico; 2Clinical Epidemiology Research Unit & Cochrane Mexico UNAM Center, Hospital Infantil de Mexico Federico Gomez, Mexico City 06720, Mexico; 3Departamento de Ciencias Sociales en Salud, Dirección de Investigaciones Epidemiológica y Psicosociales, Instituto Nacional de Psiquiatría Ramón de la Fuente Muñiz, Mexico City 14370, Mexico; 4Center for Research in Evaluation and Surveys, National Institute of Public Health, Cuernavaca 62100, Mexico

**Keywords:** obesity, overweight, prevention, treatment, health interventions, children

## Abstract

(1) Background: The importance of studying the health interventions used to prevent and treat overweight and obesity in school-aged children is imperative. This overview aimed to summarize systematic reviews that assess the effects of school-based, family, and mixed health interventions for preventing and treating overweight and obesity in school-aged children. (2) Methods: The Cochrane Collaboration methodology and PRISMA statement were followed. A search was conducted using terms adapted to 12 databases. Systematic reviews reporting interventions in children from six to 12 years old with an outcome related to preventing or treating obesity and overweight were included. Studies with pharmacological or surgical interventions and adolescents were excluded. (3) Results: A total of 15,226 registers were identified from databases and citation searching. Of those, ten systematic reviews published between 2013 and 2022 were included. After the overlap, 331 interventions for children between 6 and 12 years old were identified, and 61.6% involved physical activity and nutrition/diet intervention. Multicomponent intervention, combining physical activity with nutrition and behavioral change, school-based plus community-based interventions may be more effective in reducing overweight and obesity in children. (4) Conclusions: Plenty of interventions for childhood overweight and obesity aimed at prevention and treatment were identified, but there is a gap in the methodological quality preventing the establishment of a certain recommendation.

## 1. Introduction

Child overweight and obesity is a global public health issue with increasing trends. In 2016, over 340 million children and adolescents (5–19 years of age) were overweight or obese, and this figure has risen more than 10-fold [1,2]. This condition presents some physical and mental health complications: psychosocial (i.e., poor self-esteem, anxiety, and depression), endocrine (i.e., insulin resistance and type 2 diabetes), cardiovascular (i.e., dyslipidemia, hypertension, and endothelial dysfunction), among others [3,4,5].

Overweight and obese children often carry this condition into adulthood due to both physiological and behavioral factors [5,6]. Increased body mass index (BMI) in children is associated with an increased risk of adult morbidities [6,7], mainly for the cardiovascular system in the form of metabolic syndrome and type 2 diabetes [5,8]. Additionally, there is an increase in rates of premature death and mortality in adulthood [9].

Pediatric obesity is a multi-factor condition [3]. Some of the main controllable risk factors for this condition are dietary intake (i.e., calorie imbalance), sedentary behavior, and physical activity (PA) [10]. Socioeconomic status, sleep quality and duration, and home environment are factors that must also be addressed in prevention and treatment interventions [5,10]. Studying health interventions used to prevent and treat overweight and obesity in school-aged children is imperative.

A high volume of evidence related to children and adolescents had been identified in recent years and could represent an obstacle for decision makers [11]. Previous studies have suggested school-based interventions for the prevention of this condition [12,13], and they identified that most common treatments involved lifestyle interventions (i.e., regulated screen time for children) [14,15]. Additionally, it was reported that interventions (diet, family behavior, PA promotion, supervised exercise, lifestyle, or multicomponent interventions) were associated with a reduction in BMI [11]. Nevertheless, those previous efforts do not focus only on school-age children (6–12 years) and examine interventions that combined children and adolescents, but there are some important differences to consider between these groups. For example, it has been shown that factors such as social influence are different between these age groups, and children between 8 and 11 years old demonstrate the most susceptibility to behavioral change [16].

As a result, this review is important, because it is necessary to study the interventions used to prevent and treat overweight and obesity in school-aged children, and even though there is a high volume of evidence, there is a gap in the synthesis of information exclusive to school-aged children (6–12 years old). Conducting an OSRs with a focus on a specific age group presents the opportunity to deepen the interventions. Therefore, considering the advances that have been made in researching this topic, it is necessary to develop an OSR that identifies effective public health strategies to prevent and manage childhood obesity [17].

Thus, this overview aimed to summarize systematic reviews (SR) that assess the effects of school-based, family, and mixed health interventions for the prevention and treatment overweight and obesity in school-aged children.

## 2. Materials and Methods

This overview of systematic reviews followed the methodology proposed by the Cochrane Collaboration [18] and the Preferred Reporting Items for Systematic Review and Meta-analysis (PRISMA) [19]. The detailed protocol of the present study has been previously published [17].

### 2.1. Criteria for Considering Systematic Reviews for Inclusion

The criteria for inclusion were as follows: SR of health interventions which included studies evaluating the prevention and/or treatment of overweight and/or obesity in children with aged between 6 and 12 years or studying in the first to sixth grade of primary education. The setting of these interventions could have been schools, primary care, within the family, or mixed. The SR of randomized controlled trials (aiming to detect health interventions in these populations) or observational studies (aiming to detect public health policies applied in this population) were included. In order to be included, SRs were required to have reported upon at least one of the following outcome measures: a change in weight, BMI and/or BMI z-score, anthropometric measures (body fat percent, waist circumference [WC], waist-to-hip ratio, triceps skin-fold thickness, subscapular skin-fold thickness, etc.), cardiovascular risk factors or behaviors related to PA, and/or dietary habits and/or hydration; for either the prevention and/or the treatment of overweight and/or obesity. For a study to be considered an SR, it must have performed a comprehensive search of the literature in at least three electronic databases, a paired independent review, a critical assessment, and a risk of bias assessment. All of the relevant Cochrane and non-Cochrane SRs that matched our criteria were selected. All of the studies that did not comply with the characteristics to be considered an SR, that did not include a stratified analysis with the information of children from the established age gap, or that included pharmacological interventions were excluded from this overview. The retrieved protocols were checked for publication status, and in specific cases, authors were contacted to confirm the progress or publication status.

### 2.2. Search Methods for Identification of Reviews

Validated search strategies were used to perform the search for SRs in 12 electronic databases: PubMed, Embase, the Cochrane Library, LILACs, CINAHL, PsycINFO, PROSPERO, O.T. Seeker, TripDatabase, DARE, Epistemonikos and Health Interventions. The search was performed with no language restriction for all of the published evidence until October 2022. The descriptors included were obesity, overweight, treatment, and prevention. The results of these searches were assessed by title and abstract by two independent reviewers (LM-S and BA-N), and all of the relevant citations were retrieved for full-text assessments. The same two independent reviewers assessed the full-text articles for potential inclusion. In the case of disagreement, a third reviewer (E.D.-G.) assessed the situation.

### 2.3. Data Collection (Overlapping)

The SRs selected for inclusion were assessed for overlapping of primary studies by the creation of a reference matrix and the calculation of the corrected area (CCA) following the methods proposed by Pieper et al. [20] The followed methods ensured that no primary study outcome data were double counted which were extracted only once, and that all of the outcome data from relevant systematic reviews were included.

### 2.4. Data Extraction and Management

The data extraction was performed independently by the two reviewers in a predefined platform, and the following data were retrieved: author, year, the language of publication, date last assessed as up-to-date, objective, number of included studies, author’s information from the included primary studies, country of publication, the population included, types of studies included, SRs’ search strategies, names of databases searched in each SR; date ranges of databases searched in each SR; date of last search update in each SR; participant characteristics such as age, sex, ethnicity, stage of disease, and co-morbidities; definition of disorder; type of intervention(s); time of application, frequency, intensity and dose, the follow-up period, setting; the target population of the intervention(s); primary and secondary outcome(s); adverse events; the risk of bias of the included primary studies; quantitative outcome data; the certainty of the evidence; and limitations.

Data analysis was stratified by the given health intervention’s objective, being either prevention and/or treatment. Subgroup analysis was performed divided into the type of intervention. As our main goal was to present and describe the body of evidence currently available, all outcome data will be presented as extracted from the SRs, and no re-analysis will be performed.

Narrative summaries of findings tables are presented, and health interventions are categorized by their effectiveness or clinical importance as far as possible. As suggested in the handbook [18], only of those studies that presented a GRADE evaluation do we present a general summary of their findings.

### 2.5. Assessment of Methodological Quality of Included Systematic Reviews

The assessment of the methodological quality of the included reviews was performed independently by two reviewers using the ROBIS tool, and a summary was developed using the ROBVIS visualization tool. The three phases contemplated in the ROBIS tool (Phase 1: assessing relevance; Phase 2: identifying concerns in the review process; and Phase 3: judging risk of bias and assigning the risk of bias in the review) were assessed for each included SR using pre-formatted extraction forms, and they were presented in tables and graphics [21].

Additionally, data on the risk of bias of each primary study contained in the included SRs were extracted and presented as a summary; as we considered the possibility of different instruments having been used in the primary studies, the results are presented and summarized in a narrative form, classifying them by the type of instrument used for their assessment and the potential impact on the quality of the given SR.

## 3. Results

A total of 15,226 studies were identified from databases, registers and citation searching. Through our comprehensive database search, 11,746 records were identified, of which 271 records were duplicated. In total, 11,475 records were screened, and 10,734 records were excluded after assessment of the title and abstract; 553 full-text articles were retrieved for eligibility assessment against our criteria. From these, 539 were excluded, and 10 were selected for final inclusion. The main reason for exclusion after the full-text assessment was the included population age, followed by not being compatible with our operative definition of SRs (Figure 1). From the ten SRs included [22,23,24,25,26,27,28,29,30,31], five were focused on child overweight and obesity treatment [22,24,26,28,31], four aimed at prevention strategies [23,27,29,30], and one reported interventions for both prevention and treatment [25]. The identified overlap was CCA = 0.015, classified as slight [20].

### 3.1. Description of Included Reviews

The reviews were published between 2013 and 2022. The number of primary studies in the included reviews ranged from 4 to 146, and after the overlap between the SRs was reviewed, 331 primary studies were identified. The characteristics of the interventions from each primary article were described in Appendix A. The included population size ranged from 15 to 130,353. All of the SRs included present primary studies with interventions as randomized clinical trials (RCT). Additionally, two SRs included observational studies; Williams et al. [27] included RCT, controlled before and after studies and interrupted time series, cohort and cross sectional studies. Rochira et al. [29] included quasi-experimental, RCT, and observational studies. The age of the participants in the primary studies ranged from 4 to 12 years old; Williams et al. included studies with younger participants in the first grade of primary education [27]. The characteristics of the systematic reviews included are detailed in Table 1.

#### 3.1.1. Characteristics of the Interventions

##### Prevention

Within the five [23,25,27,29,30] SRs with the aim of prevention; the most common setting of the interventions was the school with 100%, while 80% of the SRs included family-based interventions and 50% included a community-based intervention. Of these, two [23,29] combined a school-based setting and a community-based setting. Two of the SRs [23,25] described interventions combining dietary, behavioral, and PA components. Podnar et al. [30] reported mostly PA interventions also combined with nutrition counseling or other physical fitness components, and interventions for the reduction of sedentary time. Rochira et al. [29] focused on interventions that introduce a program of school gardening. Williams et al. [27] identified diet and PA policies. The duration of the interventions had a wide heterogeneity, from 4 months to 9 years.

##### Treatment

The six [22,24,25,28,31,32] SRs that target their interventions as treatment reported family-based interventions, and all but Mead et al. had school-based interventions; two [22,28] also included interventions in health centers, while another two [22,31] included community-based interventions. All of the SRs in this category had PA and nutrition as part of their aims of intervention. For example, Albornoz-Guerrero et al. [28] identified studies that include interventions focused mainly in PA, nutrition, education and behavior, and the combinations thereof. In Jurado Castro et al. [31], 100% of the included primary studies utilized PA interventions, and some of those additionally included other interventions, such as active video games, lifestyle education and nutrition recommendations. Sbruzzi et al. [25] identified treatments using nutrition interventions, PA, education, behavioral and multicomponent interventions (interventions including those previously mentioned in addition to the promotion of healthy dietary habits and counseling for fruits and vegetables [F/V] consumption. The characteristics of the interventions identified by the systematic reviews and the main findings are described in the Table 2.

Analyzing all of the primary studies, the information could be described in terms of the type of conditions necessary for the development of the interventions, the stakeholders involved, the population to which they are targeted, and the characteristics of each intervention in the Appendix A. Of those 331 primary studies, 61.6% included diet and PA, of which 41.7% also included behavioral change intervention. Children participated in the approach in 48% of the studies, and 46.8% involved children and parent participation. In 53.4%, the educative personnel oversaw the intervention, while for 23.9% it was health professionals, and in 7.5% both were involved.

The detailed outcomes for diet or nutritional interventions, PA or exercise interventions, behavior change interventions, and multicomponent interventions for the primary studies are described in Appendix A, respectively.

### 3.2. Methodological Quality of Included Reviews

We assessed the risk of bias in the included SRs using the ROBIS tool. Globally, a 35% risk of unclear bias was observed in the reviews, mostly given by the assessment of synthesis and findings in the reviews. As two of the included reviews are Cochrane systematic reviews, a low risk regarding the first three domains was observed. Four of the included SRs do not provide information about any a priori register of their protocol [25,26,27,29]. Finally, less than 10% represent a high risk of bias. The summary and evaluation of the risk of bias are presented in the Figure 2.

Regarding the study eligibility criteria domain, five of the SRs had an unclear [27,29,30,31] or high concern [26], mainly due to not providing information about having an a priori protocol, not having specific diagnostic criteria for the condition, and the adequateness of their restrictions on eligibility criteria based on the sources of information. The identification and selection of the studies domain presented three studies with high concern [22,26,28] and one with unclear concern [31] because of the poor reporting of the rationale for search time-frames restrictions, no additional search methods other than databases, and the appropriateness of publication, language, and date restrictions. Five studies were rated with high [26,28,29] and unclear concerns [22,30], given some concerns on the efforts made to minimize errors in the risk of bias assessment and the appropriateness of the quality assessment methodology. Finally, for the synthesis and findings domain, six studies were rated as unclear concern [22,23,24,25,28,31] and one as high [26]; the main reasons for these ratings were due to concerns about the appropriateness of the synthesis given the nature and similarity in the research questions, study designs and outcomes, how the heterogeneity was addressed and the methods (or lack of them) used to demonstrate the robustness of the results.

#### Risk of Bias in the Included Systematic Reviews

The risk of bias assessment in the included SRs was predominantly (80%) evaluated following the Cochrane quality assessment tool. Andrade et al. [26] utilized the Academy of Nutrition and Dietetics’ *Evidence Analysis Manual*, and Williams et al. [27] reported the Newcastle–Ottawa scale (NOS). Additionally, Podnar et al. [30] used NOS to evaluate the quasi-experimental studies. In Table 2 the main biases described by the SRs are described.

### 3.3. Effects of Interventions

Due the wide heterogeneity of the interventions, a quantitative synthesis could not be performed.

Anthropometric measures (BMI, BMI z-score, WC and percentage of overweight) were reported for all. Others outcomes identified were blood pressure (BP) [22,24,25,26], biochemical measures [25,26], and change in dietary habits [22,25,29].

#### 3.3.1. Body Mass Index

Six reviews identified slight improvements in BMI z-score [24,26,28,29,30,31]. Andrade et al. [26] and Podnar et al. [30] identified that school-based nutrition intervention programs using community-based framework elements were more likely to elicit improvements in BMI. Mead et al. [24] and Albornoz–Guerrero et al. [28] observed that multicomponent interventions (PA, diet, nutrition education and behavior therapy) may be beneficial in achieving small, short-term reductions in BMI, BMI z-score and weight in 6–11-year-oldchildren. Jurado–Castro et al. [31] identified a reduction of BMI z-score in 53 participants under PA intervention vs. no intervention. Three SRs and meta-analyses [23,24,25] identified high heterogeneity and low certainty of evidence for BMI outcome. Rochira et al. [29] identified a significant reduction of WC and BMI% when comparing pre- and post-intervention measurements. Williams et al. [27] identified that the pool results for the School Breakfast Program had a significant lower BMI-SDS but with significant heterogeneity.

Jull et al. [22] identified no significant difference in BMI z-score in 102 participants, no significant difference in % of overweight at 6 months of follow-up in 27 participants, but with differential loss to follow-up (72% under parent-only conditions vs. 35% under parent–child conditions), and their evidence suggests that parent-only interventions might have a similar effect to parent–child interventions; however, further study is needed due to the risk of bias and the quality of evidence. Sbruzzi et al. [25] suggested that educational interventions targeted at prevention, when compared to usual care or no interventions, yielded a non-significant reduction and high rates of heterogeneity in WC, BMI, and BMI z-score. Further, in this review it was identified that educational interventions to treat child obesity, when compared to usual care or no intervention, resulted in a reduction of anthropometric measures and diastolic B.P. Loveman et al. [23]. described that parent-only interventions may be effective for treating childhood overweight when compared with wait-list control, but this has similar effects to those of parent–child interventions, and it is important to note that they reported with low quality of evidence and great heterogeneity.

Williams et al. [27] studied diet-related policies; pooled results for the National School Lunch Program presented a non-significant rise in BMI, while pooled results for other diet-related policies yielded non-significant reduction of BMI. Also, PA-related policies resulted in non-significant reduction of BMI, and combined policies had significant heterogeneity, so effects were not combined.

#### 3.3.2. Other Outcomes

Andrade et al. [26] reported six studies with an increase in F/V intake; and decrease in total calories, fats, and sodium consumptions. Also, involving parents/community improves children’s dietary behaviors, combined elements from home and school environments are key to interventions success. Ecological or community-based participatory research frameworks could be useful in programs aiming to reduce obesity-related outcomes in school-aged Hispanic children. Rochira et al. [29] identified a general improvement using school gardening interventions in F/V consumption, knowledge and nutritional behavior. This review identified a positive impact in nutritional attitudes (i.e., “Willingness to try new F/V”). Jurado-Castro et al. [31] identified that PA intervention increased moderate and vigorous PA time and engagement in intervention groups compared with the control groups in the RCTs.

## 4. Discussion

The evidence suggests that multicomponent intervention combining PA with nutrition and behavioral change may be more effective in reducing overweight and obesity in children. This report summarized the evidence related to the prevention and treatment of obesity and overweight in children between 6 and 12 years old from 10 systematic reviews. Nine of the included studies reported multicomponent intervention, while only Rochira et al. [29] focused on gardening interventions. After the overlap process, 331 interventions for children aged between 6 and 12 years were identified, and 61.6% of interventions involved PA and nutrition/diet intervention. Nevertheless, this information could not be pooled for a deep analysis of the outcome owing to the heterogeneity between the interventions.

In general, school was the most common setting for the interventions; Podnar et al. [30], the largest systematic review included, identified that school-based interventions appear to be an effective strategy in the primary prevention of childhood obesity. Furthermore, three [22,26,29] included systematic reviews reported that combined settings, such as community (i.e., parents, community centers, other teachers, restaurants or vendors) and school, could be more effective in this age group. School is obligatory almost worldwide, and it is a common location that children at this age attend. Previous studies and frameworks with interventions including this age group suggested the incorporation of more levels to promote changes in children. Social ecological frameworks in a previous review proposed focusing the interventions not only on the children but also involving other levels, such as parents and the community [33].

In accordance with previous similar overviews, the participation of the community, parents, and school was an enabler for the intervention [12]. Our analysis similarly identified the importance of school-based interventions composed of PA and nutrition/diet. For treatment, the importance of a multidisciplinary and multilevel team is noted by the Academy of Nutrition and Dietetics [15]; in the 331 interventions, we identified that just 23.9% involved health professionals, highlighting an important gap of participation. Other studies that include school-aged children and adolescents have identified similar changes in BMI, body weight and adiposity with mixed interventions for the treatment or multicomponent behavior changing [11,13,14]. Amini et al. [13] suggest that differences must be considered by sex, psychological and cultural aspects.

There are controllable factors that have been recognized to explain the etiology of obesity in children. Environmental factors such as school policies, school ambience, and demographics play an influential role in eating and activity behaviors in children [5,6]. Also, the relationships with their parents and their community are key to developing preferences and can overcome a dislike of foods [4,6]. As suggested in this review, the interventions for prevention and treatment should consider a school setting and involve parents and the community.

A strength of this overview was the analysis of quality using the ROBIS tool. Regarding the quality of the evidence identified, there was a significant body of low-quality evidence of multicomponent interventions for prevention and treatment. The leading causes of low-quality ratings are methodological flaws at the primary study and SR levels. Most of these could be prevented through simple strategies such as a priori development and registration of SR protocols, using the correct reporting guidelines for the study design, promoting the use of standardized reporting for nutrition and the prescription of exercise interventions, and strict adherence to recommended methodologies for developing SRs. This study also analyzed the overlapping of the primary studies, and the characteristics of the interventions from each primary article were included. A limitation of the OoSR methodology is that its development is currently vague, even though there have been efforts to clarify it [34].

The SRs made by Mead et al. [24], Sbruzzi et al. [25], and Loveman et al. [23] included interventions with high diversity in their analysis, a situation that alters the clinical interpretation of the values derived from their analysis. For this reason, the data could not be reanalyzed, because even though the outcomes measured and compared are the same, the interventions’ heterogeneity is significant.

None of the systematic reviews with mHealth interventions matched this study’s age group. We encourage future research to identify these kinds of interventions for overweight and obesity in this age group.

## 5. Conclusions

In this overview plenty of interventions were presented for the prevention and treatment of overweight and obesity in school-age children. Evidence suggests that multicomponent interventions that combine PA, nutrition and behavioral change components appear to be the most effective in preventing and reducing overweight and obesity in children. Nevertheless, there is a gap in methodological quality to establish robust recommendations in primary studies and the development of systematic reviews. There is a need for future reports and primary studies with more robust methodologies that provide future systematic reviews with more consistent evidence. This calls for the scientific community to develop preventive and curative solutions for this emergent health issue.

## Figures and Tables

**Figure 1 nutrients-15-00773-f001:**
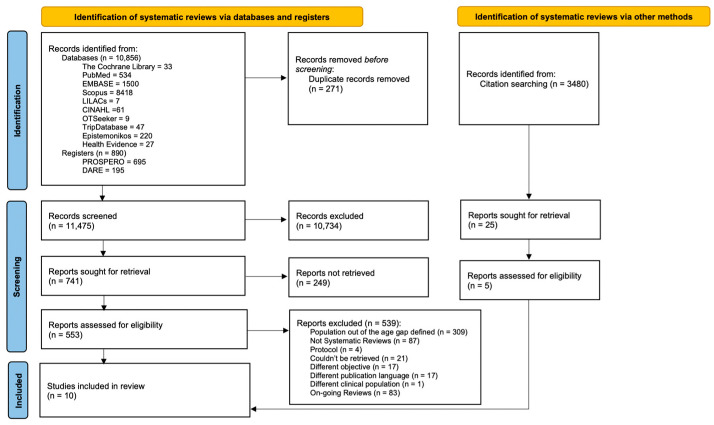
Flow chart of the included systematic reviews, following PRISMA guidelines.

**Figure 2 nutrients-15-00773-f002:**
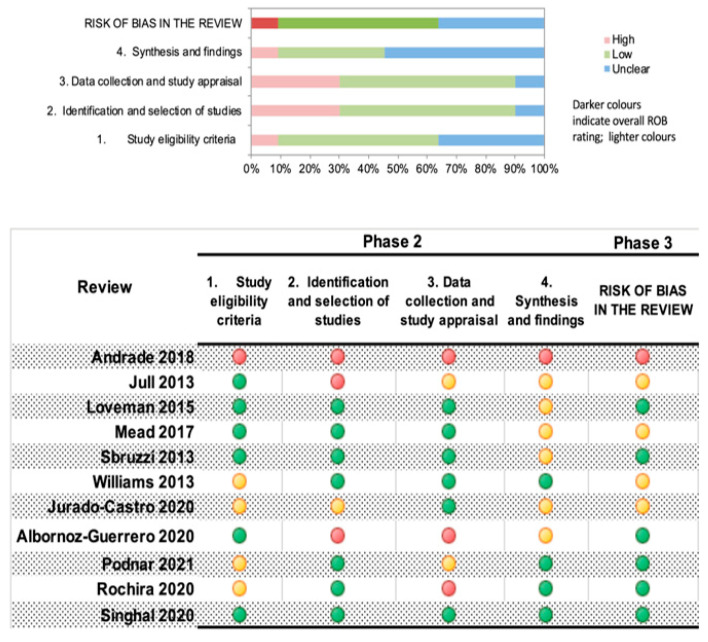
Summary and graph of risk of bias assessment, assessed with ROBIS Tool.

**Table 1 nutrients-15-00773-t001:** Characteristics of the included systematic reviews.

Author and Year	Objective	Population Gender (Age Range)	Databases Searched	Type of Included Studies	Included Studies	Total Population Included	Sample Size of the Included (Range)
Andrade et al., 2018 [26]	“Examine the frameworks used within school-based intervention programs that showed improvements in obesity-related outcomes among Hispanic children in the United States and Mexico.”	Female and male (8–10 years)	PubMed, PsycINFO, Scopus and Web Science	RCT, quasi-experimental, before and after designs	10	10,365	96–3032
Jull et al., 2013 [22]	“Assess the effectiveness of weight loss interventions that compared a parent-only condition with a parent–child condition in overweight and obese children.”	Female and male (8.7–11.2 years)	Cochrane Controlled Trials Register, Medline, Embase, PsycInfo and CINAHL	RCT	4	266	37–80
Loveman et al., 2015 [23]	“Assess the efficacy of diet, physical activity and behavioral interventions delivered to parents only for the treatment of overweight.”	Female and male (5–11 years)	The Cochrane Library, MEDLINE and MEDLINE in press, EMBASE, PsycINFO, CINAHL, LILACS, ClinicalTrials.gov, WHO ^1^ ICTRP	RCT	21	NR	15–645
Mead et al., 2017 [24]	“Assess the effects of diet, physical activity, and behavioral interventions (behavior-changing interventions) for the treatment of overweight or obese children aged 6 to 11 years.”	Female and male (6.2–11.9 years)	Cochrane Controlled Trials Register, Medline Ovid, Epub, PsycINFO, CINAHL, LILACS, ClinicalTrials.gov, WHO ICTRP	RCT, cluster RCT, parallel RCT (cross-over design)	76	8461	16–686
Sbruzzi et al., 2013 [25]	“Review educational intervention, including behavioral modification, nutrition and physical activity, as compared to usual care or no intervention, for the prevention or treatment of childhood obesity in school children aged 6 to 12 years.”	Female and male (6–12 years)	MEDLINE (PubMed), Cochrane Controlled Trials Register, EMBASE	RCT	26	23,617	70–4019
Williams et al., 2013 [27]	“Evaluate the effects of policies related to diet and physical activity in schools, either alone or as part of an intervention program on the weight status of children aged 4 to 11 years.”	Female and male (4–12 years)	Medline In-Process and Other Non-Indexed Citations [Ovid], Medline [Ovid], EMBASE [Ovid], PsycINFO [Ovid], SportDISCUS [Ebscohost], Web of Science [ISI Web of Knowledge], Education Resource Information Center (ERIC) [Dialog Datastar], British Education Index [Dialog Datastar], Australian Education Index [Dialog Datastar], Cumulative Index to Nursing and Allied Health Library (CINAHL Plus) [Ebscohost], and The Cochrane Library [Wiley Online]. metaRegister of Controlled Trials, Clinical Trials.gov and the International Clinical Trials Registry Platform	RCT, controlled before and after studies and interrupted time series, cohort, and cross sectional studies	21	194,358 * (approx.)	34–130,353
Albornoz-Guerrero et al., 2021 [28]	“Analyze the characteristics of multicomponent interventions to reduce childhood overweight and obesity in territories with an extremely cold climate.”	Male–female (6–12 years)	Medline, PubMed, PsycNet, SciELO, grey literature.	RCT	29	4434	16–685
Jurado-Castro et al., 2020 [31]	“Measure the effects of current interventions with a physical activity component on the body mass index (BMI) Z-score and on the moderate and vigorous physical activity (MVPA) time, measured by accelerometry, and focused on children with obesity.”	Male–female (6–12 years)	MEDLINE (PubMed), Cochrane Register of Controlled Trials (CENTRAL), Web of Science, ScienceDirect (SCOPUS), PROQuest, BVS (Biblioteca Virtual en Salud), Annual Reviews, LILACS (Literatura Latino Americana y del Caribe en CC de la Salud), Dialnet, Scielo.	RCT	10	952	26–322
Podnar et al., 2020 [30]	“Compare the effects of interventions that targeted sedentary behaviours or physical activity (PA) or physical fitness on the primary prevention of obesity in 6- to 12-year-old children.”	Male–female (5.5–12.49 years)	MEDLINE, The Cochrane Central Register of Controlled Trials (CENTRAL), Scopus, LILACS, OpenGrey, Open Access Thesis and Dissertations, Clinical Trials, WHO International Clinical Trials.	RCT, quasi-experimental	146	NR	75–2682
Rochira et al., 2020 [29]	“Analyze the main elements of school gardening with a specific meta-analysis about its impact on anthropometric parameters.”	Male–female	PubMed, EMBASE, and Cochrane Library	Quasi-experimental, RCT, observational.	33	NR	30–3769

^1^ WHO ICTRP: World Health Organization International Clinical Trials Registry Platform; NR: no report; RCT: randomized clinical trial. * The population was not reported in all the primary studies.

**Table 2 nutrients-15-00773-t002:** Characteristics of the interventions and main findings and risk of bias assessment of the included systematics reviews.

Author and Year	Aims of the Intervention	Intervention Setting	Main Outcomes	Main Findings	Risk of Bias Tool Key Points
Treatment					
Albornoz-Guerrero et al., 2021 [28]	Physical activity Nutrition Education Behavioral	Family-based, school-based, health centers	Nutritional status: BMI, BMI z-score, WC, body composition. Physical and health condition: Physical activity, food intake, blood pressure, health biomarkers. Psychological variables: Health-related quality of life.	Interventions were effective when components of physical activity, diet, education, and behavioral therapy were combined.	Cochrane Risk of Bias (ROB-tool) 22% high risk of bias for performance bias. Low risk or unclear risk of bias in other domains.
Jurado-Castro et al., 2020 [31]	Physical activity and active video games Lifestyle education Recommendations of nutrition	Family-based, school-based and community-based	BMI z-score Physical activity	Interventions with physical activity seem to be successful in reducing BMI and increase time spent engaged in PA.	Cochrane Risk of Bias tool (ROB-tool) 60% High risk of bias for other bias. 10% Unclear risk of bias for blinding of participants, personnel, and outcome data.
Andrade et al., 2018 [26]	Nutrition education Physical education Physical activity	Family-based, school-based	Change in dietary habits BMI Blood pressure	School-based nutrition intervention programs with elements of community-based framework were more likely to elicit improvements in BMI of this population.	AND Evidence Analysis Manual 60% studies high quality 40% studies neutral quality
Jull et al., 2013 [22]	Promotion of healthy dietary habits Physical activity Behavioral approach	Family-based, school-based, health centers, community-based	BMI BMI z-score BMI standard deviation	Parent-only intervention had a similar effect to parent–child interventions for weight loss.	Cochrane Tool for Quality Assessment Overall risk of bias was unclear or high. 75% High risk of bias in blinding and incomplete outcome
Mead et al., 2017 [24]	Behavioral approach Diet Physical Activity	Family-based	BMI BMI z-score Weight	Multicomponent behavior-changing interventions that incorporate diet, physical activity and behavioral change components may be beneficial in achieving small, short-term reductions in BMI; BMI z-score and weight.	Cochrane Risk of Bias (ROB-tool) Low Risk of Bias: 75% low risk of bias in random sequence generation, allocation concealment. High risk of bias: >50% blinding of participants and personnel (objective outcomes), (subjective outcomes). GRADE Low quality, for BMI, BMI z-score, weight, adverse events, and parent-reported health-related quality of life outcomes Very low for child reported HRQoL Downgrade due to risk of bias, inconsistency, and imprecision
Sbruzzi et al., 2013 ^ß^ [25]	Prevention Behavioral approach Nutrition Physical activity Education Treatment Behavioral approach Nutrition Physical activity Education	Family-based, school-based	WC BMI BMI z-score SBP DBP Total cholesterol HDL.C	Educational interventions are effective for treating obesity and its consequences but not for prevention. Due to low quality and high heterogeneity among studies, trials with more comprehensive and specific strategies are needed.	Cochrane Tool for Quality Assessment 83.3% Do not report or report unclear in allocation concealment. 11% report blinding of outcome assessors. GRADE Very low quality for BMI z-score, WC, BMI, DBP, total cholesterol and HDL-C. Low quality for systolic blood pressure. Downgraded due high heterogeneity, imprecision, limitations in design.
Prevention					
Williams et al., 2013 [27]	School policy for: Physical activity Diet Both	School-based	BMI BMI standard deviations BMI percentiles BMIHFZ	SBP was associated with a significant decrease in BMI-SDS. PA policies were not associated with significant changes. Diet and physical policies need to be located within more complex approaches in order to prevent childhood obesity.	Newcastle-Ottawa scale (NOS) High lost to follow up rate Samples were representative of the population Not reported a general or overall quality of the studies.
Loveman et al., 2015 [23]	Behavioral approach Diet Physical Activity	School-based, health centers, university-based and community	BMI Body weight	Parent-only interventions are similar to parent–child interventions and minimal contact interventions. There was a difference with the waitlist children. Also, it is important to consider the sample sizes of many trials, the loss to follow-up and the low quality of evidence.	Cochrane Risk of Bias tool High and unclear risk of bias. 50%. High risk of bias incomplete outcome data (objective outcomes) and selective reporting. GRADE evaluation Overall low quality, downgrade due to risk of bias (attrition), small number of trials and sample sizes.
Podnar et al., 2020 [30]	Physical activity Physical fitness Reduce sedentary behavior	School-based	BMI BMI z-score %BF	School-based PA interventions could be effective in the prevention of obesity. Interventions that combined PA or fitness components with strategies to reduce sedentary behavior were less effective.	Risk of Bias tool (ROB-tool) 26.4% of RCT had an overall low risk of bias. 67% high or unclear risk relating to low intervention fidelity. Newcastle-Ottawa scale (NOS) 21.8% of non-randomized studies had 6/8 points, considered overall low risk.
Rochira et al., 2020 [29]	School gardening	School-based, community based, family based.	F/V consumption Anthropometrics: BMI, BMI z-score, waist circumference. Other outcomes: Blood pressure, urinary samples, blood samples	School gardening had an increase in F/V daily/weekly intake and improved their knowledge on this topic. Modest but clinically significant reduction of WC and BMI%.	Cochrane Tool for Quality Assessment and Strengthening the Reporting of Observational Studies in Epidemiology (STROBE) 20% Good quality 80% Fair quality

BMI: Body mass index; WC: waist circumference; PA: physical activity; %BF: body fat percentage; SBP: systolic blood pressure; DBP: diastolic blood pressure; F/V: fruits and vegetables; BMIHFZ: healthy fitness zone BMI categorization; BMI SDS: BMI standard deviations; HRQoL: hazard ratio quality of life; AND: Academy of Nutrition and Dietetics. ^ß^ Sbruzzi et al., present interventions for treatment and prevention of obesity.

## Data Availability

The data that support the findings of this study are available from the corresponding author (E.D.-G.) upon reasonable request.

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
