# Peer review of "Overview of Systematic Reviews of Health Interventions for the Prevention and Treatment of Overweight and Obesity in Children"

_nutrients, 2023, doi:10.3390/nu15030773_

Round 1
Reviewer 1 Report
The authors aimed to summarize systematic reviews that assess the effects of school-based, family, and mixed health interventions for the prevention and treatment of overweight and obesity in school-aged children. They followed Cochrane Collaboration methodology and PRISMA statement. Systematic reviews, reporting interventions in children from 6- 12 years old with an outcome related to prevent or treat obesity and overweight were included from 12 databases. 10 systematic reviews were included. After the overlap, 331 interventions were identified. 61.6% interventions involve physical activity and nutrition/diet intervention. Based on their results, multicomponent intervention, combining physical activity with nutrition and behavioral change, school-based plus community-based interventions may be more effective to reduce overweight and obesity in children. They concluded that several interventions for children overweight and obesity aimed to prevent, and treat were identified but there is a gap of methodological quality to establish a certain recommendation.
This is an excellent review focusing on the gap of methodological quality of former systematic reviews.
Comments
1. Formatting of tables is needed.
2. Figure 2. is probably the most important part of the manuscript. The figure could be larger. The content could be more extensively discussed to highlight the possible methodological bias.
3. Conclusions could be more direct to improve the quality of future research works.
4. English needs some editing.
Author Response
- Formatting of tables is needed.
Thank you for the observation, the tables had been modified and formatted according to the journal criteria. Please see changes accordingly.
- Figure 2. is probably the most important part of the manuscript. The figure could be larger. The content could be more extensively discussed to highlight the possible methodological bias.
Thank you for the comment, we add detailed information in Methods section, we improve this in the results section and the discussion. Please see changes accordingly.
- Conclusions could be more direct to improve the quality of future research works.
Thank you for your remark, we rewrite the conclusion adding more information about this. Please see changes accordingly.
- English needs some editing.
Thank you for the suggestion, a native English speaker checks the manuscript and some editions have been done.
Reviewer 2 Report
Dear Authors,
Thank you for your manuscript.
Please see my comments below.
In the abstract, please indicate the period of publishing of the studies included in the manuscript.
In the Introduction, the importance of this paper is not well explained.
Also, there is a disagreement between the inclusion criteria provided in the Methods section (6-12 years, line 71) and the Results (4-12 years, line 161).
The description of the quality assessment of the studies included should be more detailed (section 2.5.).
The results and the discussion sections should be better structured, providing information separately on family-based, school-based and community-based interventions and focusing more on their strengths and limitations.
Author Response
- In the abstract, please indicate the period of publishing of the studies included in the manuscript.
Thank you for the suggestion, we added that information to the abstract. Please see changes accordingly.
- In the Introduction, the importance of this paper is not well explained.
Thank you for your observation that give us the opportunity to explain better this point. We clarify this in the introduction adding a paragraph of the importance please see changes accordingly
- Also, there is a disagreement between the inclusion criteria provided in the Methods section (6-12 years, line 71) and the Results (4-12 years, line 161).
Thank you for the remark. For our study was important to identify the systematic reviews of school-age children, this includes children studying first to sixth grade of primary education. We made changes in the methodology and clarify the systematic review that include younger kids with the reason of inclusion in results.
- The description of the quality assessment of the studies included should be more detailed (section 2.5.).
Thank you for the suggestion, we deepen in that area of our review adding information in methos, results and conclusions. Please see changes accordingly
- The results and the discussion sections should be better structured, providing information separately on family-based, school-based and community-based interventions and focusing more on their strengths and limitations.
Thank you for the observation, the nature of the overview is to analyze the synthesis provide in the systematic review included. We do not have enough data from the systematic reviews to provide information separately on the setting. Although, we understand the importance of the context in this intervention, so a part of the discussion was rewritten and a paragraph were added focusing on this.